

# Central Angiotensin II type 1 receptor deficiency alleviates renal fibrosis by reducing sympathetic nerve discharge in nephrotoxic folic acid–induced chronic kidney disease

Qijun Wan[1,2,*], Zhichen Yang[3,*], Lingzhi Li[1,2] and Liling Wu[1,2]

[1] Nephrology, Shenzhen Second People's Hospital, Shenzhen, Guangdong, China
[2] Nephrology, The First Affiliated Hospital of Shenzhen University, Shenzhen, Guangdong, China
[3] Nephrology, The Third Affiliated Hospital of Southern Medical University, Guangzhou, Guangdong, China
[*] These authors contributed equally to this work.

Corresponding author
Liling Wu, liling156230@163.com

## ABSTRACT

**Background**. Fibrosis after nephrotoxic injury is common. Activation of the paraventricular nucleus (PVN) renin-angiotensin system (RAS) and sympathetic nervous system (SNS) are common mechanism of renal fibrosis. However, there have limited knowledge about which brain regions are most affected by Angiotensin II (Ang II) after nephrotoxic injury, what role does Angiotensin II type 1a receptors (AT1R) signaling play and how this affects the outcomes of the kidneys.

**Methods**. In nephrotoxic folic acid–induced chronic kidney disease (FA-CKD) mouse models, we have integrated retrograde tracer techniques with studies on AT1afl/fl mice to pinpoint an excessively active central pathway that connects the paraventricular nucleus (PVN) to the rostral ventrolateral medulla (RVLM). This pathway plays a pivotal role in determining the kidney's fibrotic response following injury induced by folic acid.

**Results**. FA-CKD (*vs* sham) had increased in the kidney SNS activity and Ang II expression in the central PVN. The activation of Ang II in the PVN triggers the activation of the PVN-RVLM pathway, amplifies SNS output, thus facilitating fibrosis development in FA-CKD mouse. Blocking sympathetic traffic or deleting AT1a in the PVN alleviated renal fibrosis in FA-CKD mice.

**Conclusions**. The FA-CKD mice have increased the expression of Ang II in PVN, thereby activating AT1a-positive PVN neurons project to the RVLM, where SNS activity is engaged to initiate fibrotic processes. The Ang II in PVN may contribute to the development of kidney fibrosis after nephrotoxic folic acid-induced kidney injury.

## INTRODUCTION

Nephrotoxic acute renal failure is a common occurrence among hospitalized patients (*Ortega et al., 2005*; *Sales & Foresto, 2020*). This condition remains a substantial contributor

to the advancement of end-stage kidney disease (*Stallons, Whitaker & Schnellmann, 2014*). The transition of chronic kidney disease (CKD) towards end-stage renal failure is intricately tied to the presence of tissue scarring or fibrosis (*Wen et al., 2019*; *Xavier et al., 2015*). The underlying mechanisms that facilitate renal fibrosis after drug toxic challenge remain to be elucidated.

Following kidney injury, the activation of the sympathetic nervous system significantly contributes to the promotion of renal fibrosis (*Cao et al., 2017*; *Cao et al., 2015*). The regulation of sympathetic nerve activity (SNA) is primarily governed by crucial regions, notably the paraventricular nucleus (PVN) of the hypothalamus, serving as a central regulatory center for autonomic outflow (*Guyenet, 2006*; *Molinas et al., 2023*). The PVN sends efferent projections to the rostral ventral lateral medulla (RVLM) in the lower brainstem and spinal cord (*Ciriello & Calaresu, 1980*; *Koba et al., 2018*; *Solano-Flores, Rosas-Arellano & Ciriello, 1997*). This complex process is crucial for regulating peripheral sympathetic discharge (*Solano-Flores, Rosas-Arellano & Ciriello, 1997*). Nevertheless, the specific role of the PVN and sympathetic nervous system (SNS) in models of nephrotoxicity-induced renal fibrosis remains unclear.

Brain renin-angiotensin system (RAS) has been implicated in the regulation of SNA (*Zucker, Xiao & Haack, 2014*). Infusion of angiotensin II (Ang II) directly into the brain has been shown to stimulate sympathetic outflow (*Campese, Shaohua & Huiquin, 2005*; *Pellegrino et al., 2016*). The PVN neurons express high levels of angiotensin II type 1a receptors (AT1a) (*Paul, Poyan Mehr & Kreutz, 2006*). Prior research has shown that the activation of AT1aR signaling in the PVN correlates with kidney injury induced by 5/6-nephrectomy, which parallels increasing sympathetic outflow (*Cao et al., 2015*). However, the involvement of RAS in the PVN in nephrotoxic-induced kidney injury has not yet been established.

This study reveals a neural mechanism by which Ang II signals originating from the PVN activate the SNS following kidney injury induced by folic acid. Furthermore, we employ virus tracing to demonstrate the involvement of a central pathway connecting the PVN and RVLM in the heightened sympathetic discharge that contributes to fibrosis. These findings shed light on the previously unrecognized role of the PVN-renal axis in regulating folic acid-induced kidney fibrosis.

# MATERIAL AND METHODS

## Experimental animals

For the animal experiments, male mice aged 8–10 weeks and weighing 20–24 g were employed. Wildtype controls consisted of age- and sex-matched C57BL/6J mice obtained from the same vendor. The mice were maintained in specific pathogen-free (SPF) conditions with unrestricted access to food and water. They were then randomly allocated into four groups: sham, folic acid nephropathy at 7 days, 14 days, and 28 days groups. Male C57BL/6J mice were administered a single intraperitoneal injection of folic acid (Sigma-Aldrich, St. Louis, MO, USA) at a dosage of 250 mg/kg, dissolved in a 0.3 mol/L sodium bicarbonate solution. They were euthanized after 7, 14, or 28 days (*Martin-Sanchez et al., 2017*).

Control mice was administered a single intraperitoneal injection of sodium bicarbonate. Each group consisted of six mice housed in animal cages, totaling approximately 72 mice used in the study. The sample sizes were determined based on our previous experiments. Euthanasia was performed by cervical dislocation following intraperitoneal injection of an overdose of 1.5% pentobarbital sodium anesthetic (100 mg/ml). Humane endpoints were established, and animals were euthanized before the planned end of the experiment if they exhibited one of the following conditions: rapid weight loss of 15–20%, persistent hypothermia, or clear signs of imminent death. Mice that were euthanized prematurely or exhibited large individual differences were excluded from the statistical analysis. All experimental procedures were approved by animal Ethics Committee of Nanfang Hospital (NFYY-2019-0135) and the Animal Ethics Committee of Shenzhen Zhongke Industrial Holdings Co. (20240015), and were conducted in compliance with their guidelines.

The Agtr1a-floxed mice (AT1a$^{fl/fl}$, CKOCMP-21237-Agtr1a) with C57BL/6J genetic background were procured from Cyagen Bioscience (Guangzhou, China).

To block sympathetic traffic, groups of mice with FA-CKD underwent total kidney denervation (*Cao et al., 2017*). To delete AT1$_a$ in the PVN, AAV-hsyn-Cre (Hanbio Biotechnology, Cat# HBAAV-3017) was injected into the PVN of AT1a$^{fl/fl}$ mice.

**Brain surgery and viral injections**

Mice were anesthetized with intraperitoneal injection of sodium pentobarbital and then positioned within a stereotactic apparatus (Stoelting, IL, USA). Following the creation of a small craniotomy hole, virus injections were administered using a micropipette connected to a Nanoliter Injector (NANOLITER 2020; WPI, Sarasota, FL, USA) and its controller (Micro4, WPI). To selectively delete AT1$_a$ in the PVN, bilateral injections of AAV-hSyn-Cre (virus titers:5.24 $10^{12}$ vector genomes /ml; 0.3 uL/injection) were performed in the PVN of AT1a$^{fl/fl}$ mice. For retrograde labeling with dyes, a glass needle loaded with PBS containing 1.0 mg/mL of CTb-555 was placed at the RVLM (*Nomura et al., 2019*). All AAVs (Serotype 2/9) and rAAVs (Serotype 2/retro) were sourced from Brain VTA Co., Ltd. Injection coordinates were referenced to bregma, as per the Paxios and Franklin Mouse Brain Atlas. The coordinates for PVN were: anteroposterior, −0.8 mm; lateral, G0.2 mm; ventral, −5.0 mm. The coordinates for RVLM were: anteroposterior, −6.8 mm; lateral, +1.1 mm; ventral, −5.9 mm. Immunostaining was conducted to confirm all injection sites (*Su et al., 2023*).

**Recording sympathetic nerve activity**

For assessing kidney sympathetic nerve activity, mice were anesthetized with urethane. The kidney nerve fibers were isolated. The nerves near the incision were then placed onto a pair of platinum electrodes (Unique Medical) to record sympathetic nerve activity. The nerve signal was amplified 10,000 times using an ERS 100C amplifier and filtered with a band-pass between 1 and 3,000 Hz. Subsequently, the nerve signal was collected and processed simultaneously by a recording system (Acqknowledge, Biopac System, CA, USA), and then stored for offline analysis. At the conclusion of each experiment, euthanasia was performed; any residual electrical activity was considered background

noise and subtracted from the total activity recorded during the experiment to obtain an estimate of the true neural activity. Nerve action potentials were quantified using a spike discriminator (Acqknowledge, Biopac System, CA, USA), with the threshold level set slightly above the noise level and kept constant throughout the experiment (*Su et al., 2023*).

### Measurement of kidney norepinephrine (NE) and Ang II

Kidney tissue NE concentrations were quantified using an ELISA kit (Demeditec Diagnostics, Kiel, Germany).

After the mice were euthanized, blood was collected, and physiological saline was perfused. The whole brain was placed in dry ice to cool. On the day of ANGII detection, the paraventricular nucleus was dissected from flash-frozen brains in the chamber of a cryostat (at −15 °C). The brains were placed in a cooled metal brain block, and remove the PVN tissue using the blade and the anatomical location of the PVN. Specifically, the cross-sectional area of the PVN is typically between 0.2 and 0.5 square millimeters, while the volume generally falls between 0.5 and 1 cubic millimeter. The tissue Ang II levels were determined *via* liquid chromatography/mass spectrometry (*Ali et al., 2014*).

### Kidney histological procedures

Mice were anesthetized, and their kidneys were excised, embedded in paraffin, and sliced at a thickness of 3 um. Kidney fibrosis was assessed by stained with Masson's trichrome and fibrosis was quantified using the NIH ImageJ program (*Cui et al., 2018*).

### In situ hybridization

Mice brains were harvested and perfused with 4% paraformaldehyde, followed by preparation of 50 μm frozen sections. Subsequently, brain sections underwent digestion with proteinase K and hybridization with a digoxigenin-labeled LNATM Detection Probe for AT1a (Qiagen, Hilden, Germany) at 54 °C for over 16 h. The sections were incubated with peroxidase-conjugated sheep anti-digoxigenin (1:5000, 11-207-733-910, Roche, Basel, Switzerland) at 4 °C for 16 h, and then subjected to detection using the TSA-Plus Fluorescence System (APExBIO, TX, USA). The probe sequence for AT1a was /5DiGN/ATGTCACGGTTGGTACAAGCA/3DiG _N/.

### Immunostaining

Immunostaining procedures followed previously published methods (*Cao et al., 2017*). Briefly, paraffin sections (3 um thick) and frozen brain sections (50 um thick) were incubated with primary antibodies at 4 °C for 24 h, followed by incubation with corresponding secondary antibodies at 4 °C for another 24 h. Afterward, the sections were mounted onto glass slides and observed using a Leica confocal microscope (Leica TCS SP2 AOBS; Leica, Buffalo Grove, IL, USA). The primary antibodies used included: anti-Cre (1:1000, ab190177), anti-Ang II (1:800, T-4007, Peninsula Laboratories, CA, USA), anti-AGT (1:100, A6279; ABclonal, Wuhan, China), anti-c-fos (1:50, ab208942) (all sourced from Abcam, Cambridge, UK), and anti-tyrosine hydroxylase (1:100, PB9449) (all obtained from Boster, Wuhan, China).
## Quantitative PCR (qPCR)

Total RNA was isolated from 50 mg of each tissue sample stored at $-80\ °C$, following the protocol provided with the RNAex Pro RNA Extraction Reagent (#AG21102; Accurate Biotechnology, Shenzhen, China). RNA quality was verified using a NanoDrop One spectrophotometer (Thermo Fisher Scientific, Waltham, MA, USA), ensuring an A260/280 ratio between 1.8 and 2.0. Genomic DNA was digested, and reverse transcription was conducted using the Evo M-MLV RT Mix Kit and the gDNA Clean for qPCR kit (# AG77128; Accurate Biotechnology, Shenzhen, China). Subsequently, 1.0 $\mu$g of RNA was converted into cDNA using oligo(dT) primers in a 20 $\mu$L reaction mixture. The reverse transcription was carried out at 37 °C for 15 min, followed by 5 s at 85 °C. The cDNA was then stored at $-80\ °C$. qPCR was performed using a LightCycler 96 Real-Time PCR Detection System with a 20 $\mu$L reaction mixture containing 2 $\mu$L of cDNA, 10 $\mu$L of 2X SYBR Green Pro Taq HS Premix (#AG11701; Accurate Biotechnology, Shenzhen, China), and gene-specific primers (AXYGEN, PCR-0208-C). The PCR protocol included an initial denaturation at 95 °C for 30 s, followed by 40 cycles of 95 °C for 5 s and 60 °C for 30 s. Melting curve analysis was conducted at 95 °C for 15 s, 60 °C for 1 min, and 95 °C for 5 s, resulting in a unimodal dissolution curve. No Cq value was observed for the No Template Control (NTC). Gene-specific primers (Table S1) were designed to produce amplicons of approximately 100 bp, confirmed *via* NCBI BLAST. Fold changes in RNA levels were calculated using the $2^{(-\Delta\Delta CT)}$ method, with GAPDH as the internal reference. The slope ranged between $-3.59$ and $-3.1$, with an $R^2$ value of $\geq 0.9$. The limit of detection (LOD) was determined to be 2.5 target molecules with a 95% confidence interval. Data analysis excluded any technical replicates that significantly deviated from the other two values among the three technical replicates. Each sample was analyzed in three technical replicates to ensure reliable results.

## Quantitative and statistical analysis

Continuous variables were presented as mean $\pm$ SD. Group differences were assessed using one-way ANOVA or unpaired t-tests with Bonferroni correction for multiple comparisons (SPSS software, version 19.0; SPSS, Inc., Chicago, IL, USA). A *p*-value of less than 0.05 was considered statistically significant.

# RESULTS

## The SNS activity and Ang II expression in the PVN is increased in nephrotoxic folic acid-induced CKD mice

To investigate the underlying mechanisms of toxicity-induced chronic kidney disease, a model of CKD induced by folic acid was utilized. Figure 1 depicts a progressive increase in renal fibrosis (Figs. 1A and 1B), as evaluated through histological analysis from day 7 to day 28 post-induction. Concurrently, there was a progressive rise in both SNS activity within the kidney (Fig. 1C) and levels of NE in the kidney (Fig. 1D) over the same timeframe.

The local renin-angiotensin system was evaluated in the PV of the hypothalamus from mice with FA-CKD or sham-operated mice, from day 7 to day 28. Between days 7 and 28 post-folic acid-induced kidney injury, there was a notable increase in the abundance of

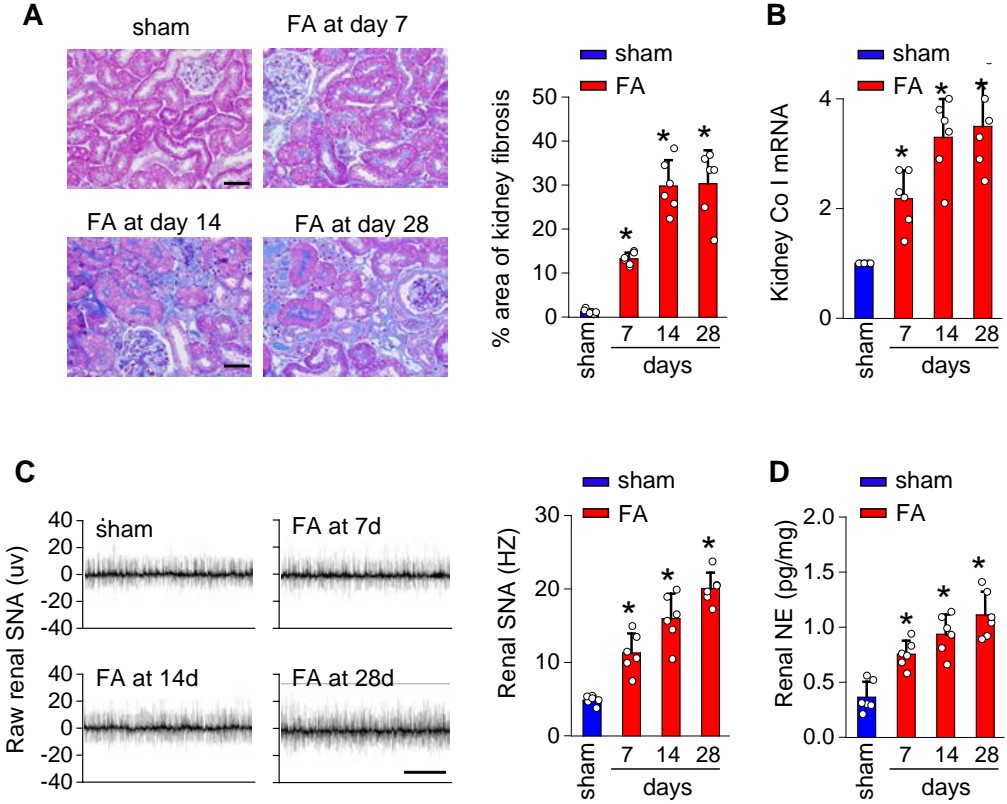

**Figure 1** **Kidney fibrosis in folic acid-induced kidney is accompanied by enhanced sympathetic nervous system (SNS) discharge to the kidney.** (A) Kidney fibrosis was visualized by Masson trichrome staining in folic acid-induced kidney injury from mice at various time points: representative images (left) and quantitative data (right). Scale bar, 50 um. (B) Level of collagen I (Co I) mRNA in kidney homogenates of the folic acid-induced kidney injury mice. (C) Sympathetic nerve activity (SNA) in the folic acid-induced kidney: representative raw records (left) and quantitative data (right). Scale bar, 2s. (D) Levels of norepinephrine (NE) in kidney homogenates of the folic acid-induced kidney. *, $p < 0.01$ *versus* sham. Error bars, mean $\pm$ standard deviation ($n = 6$ in each group). One-way analysis of variance or t test with Bonferroni correction.

cells expressing Ang II and angiotensinogen (AGT) within the PVN (Figs. 2A and 2C). This elevation correlated with an upregulation of Ang II protein levels and AGT mRNA levels in the PVN (Figs. 2B and 2D). However, there were no observed changes in the expression of angiotensin II receptor type 1a (AT1a) in the PVN region (Fig. S1A), nor in the levels of Ang II in the plasma following folic acid-induced kidney injury (Fig. S1B). Thus, The progressive fibrosis observed in folic acid-induced kidney injury was linked to continuous upregulation of Ang II in the PVN, as well as sustained elevation in SNS activity directed towards the kidney. .

## Stimulation of AT1aR signaling in the PVN amplifies kidney SNS output, thus facilitating fibrosis in FA-CKD mice

To investigate the causal relationship between AT1aR signaling in the PVN and kidney fibrosis in FA-CKD mice, selective deletion of AT1a was performed in the PVN. This was

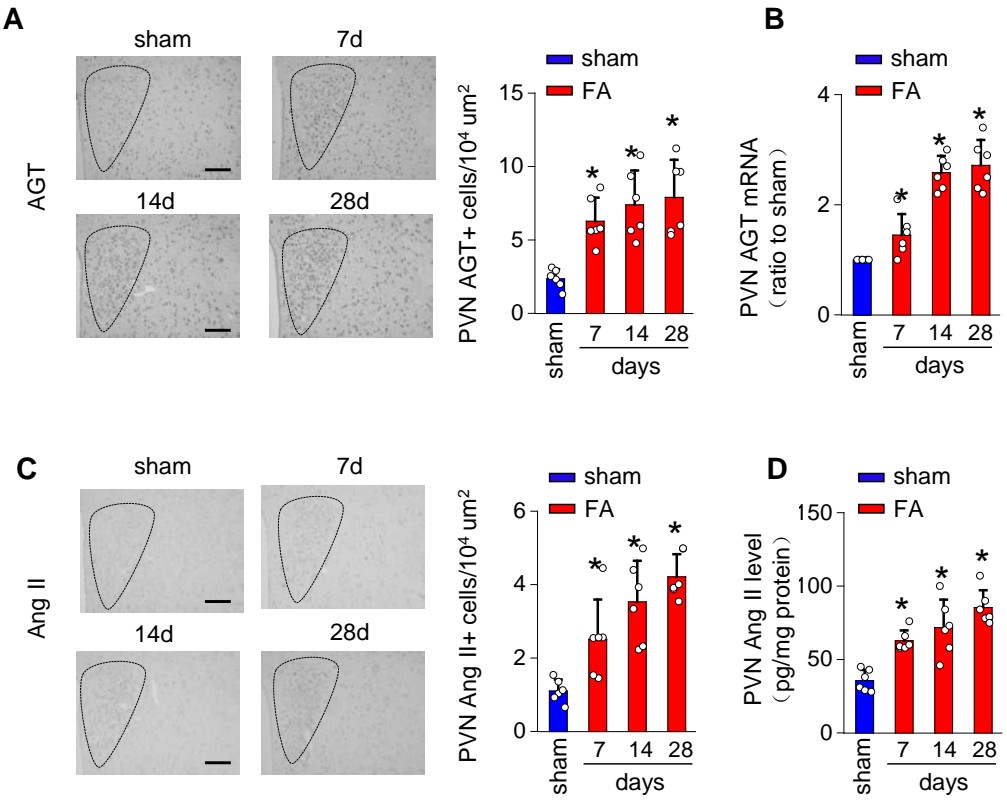

**Figure 2** **Folic acid-induced kidney injury in mice is accompanied by activity of the local renin-angiotensin system in the PVN.** (A) Immunostaining for angiotensinogen (AGT) in paraventricular nucleus (PVN, Bregma −0.94 mm): representative images (left) and quantitative data (right). Scale bar, 100 um. (B) Level of AGT mRNA in PVN homogenates. (C) Immunostaining for angiotensin II (Ang II) in PVN (Bregma −0.94 mm): representative images (left) and quantitative data (right). Scale bar, 50 um (D) Concentration of Ang II in PVN homogenates.

achieved by injecting an adeno-associated virus that encodes Cre-recombinase, which was controlled by a neuron-specific promoter (synaptophysin), into the PVN of AT1a$^{fl/fl}$ mice (AAV-hsyn-Cre; Fig. 3A). The expression of AT1a mRNA in the PVN was nearly eradicated (Figs. 3B and 3C), affirming the targeted deletion of the AT1a gene specifically in the PVN. The elimination of AT1a in PVN within the FA-CKD model resulted in a reduction in kidney SNS activity (Figs. 3D and 3E), and mitigated kidney fibrosis (Figs. 3F and 3G). Kidney denervation is produced through the blockade of sympathetic outflow. Kidney denervation was given at the same time after PVN AT1a deficiency in FA-CKD model, renal fibrosis was not further improved. Thus, the enhanced, localized Ang II in the PVN triggers fibrosis in the FA-CKD model *via* SNS activation (Figs. 4B–4D). Consequently, the increased Ang II localized in the PVN after folic acid-induced kidney injury activates the renal SNS, leading to the kidney fibrosis.

AT1aR signaling in the PVN triggers the activation of the PVN-RVLM pathway, leading to an increased SNS discharge to the kidney in FA-CKD model.

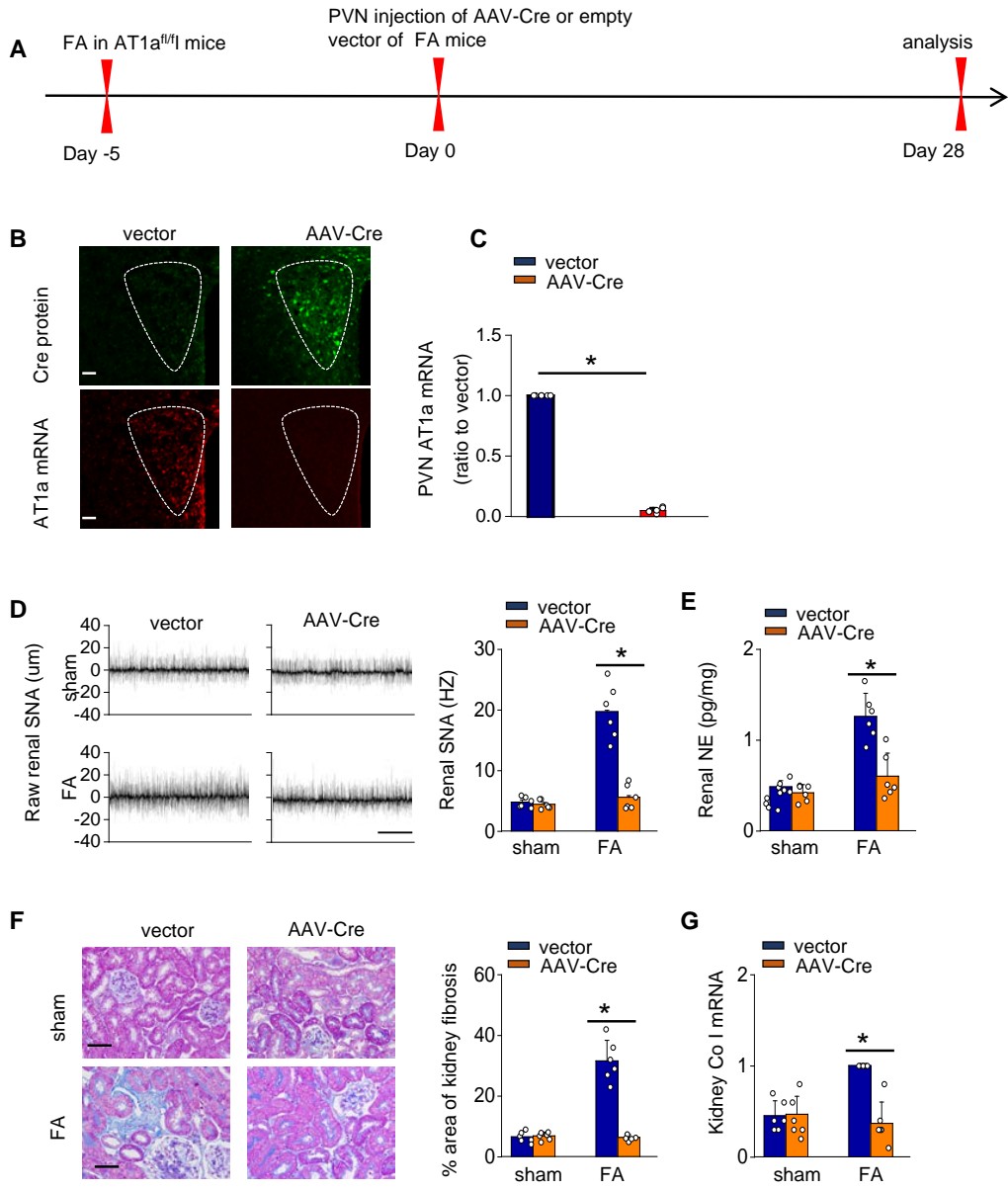

**Figure 3  Activated Ang II signaling in PVN enhances SNS discharge that promotes fibrosis in the FA-CKD mice.** (A) Outline of experimental design: PVN-specific deletion of Ang II type 1a receptor (AT1a) was achieved by injecting AAV-Cre into PVN of AT1a$^{fl/fl}$ mice. (B) Immunostaining of Cre protein and *in situ* hybridization of AT1a mRNA in PVN. Scale bar, 50 um. (C) Level of AT1a mRNA in PVN homogenates. (D) Sympathetic nerve activity (SNA) in the kidney: representative raw records (left) and quantitative data (right). Scale bar, 1 s. (E) Kidney norepinephrine (NE) level in homogenates of the FA-CKD mice. (F) Kidney fibrosis was visualized by Masson trichrome staining in FA-CKD mice: representative images (left) and quantitative data (right). Scale bar, 50 um. (G) Level of collagen I (Co I) mRNA in kidney homogenates of the FA-CKD mice. Error bars, mean G standard deviation ($n = 6$ in each group). *, $p < 0.01$. One-way analysis of variance or t test with Bonferroni correction.

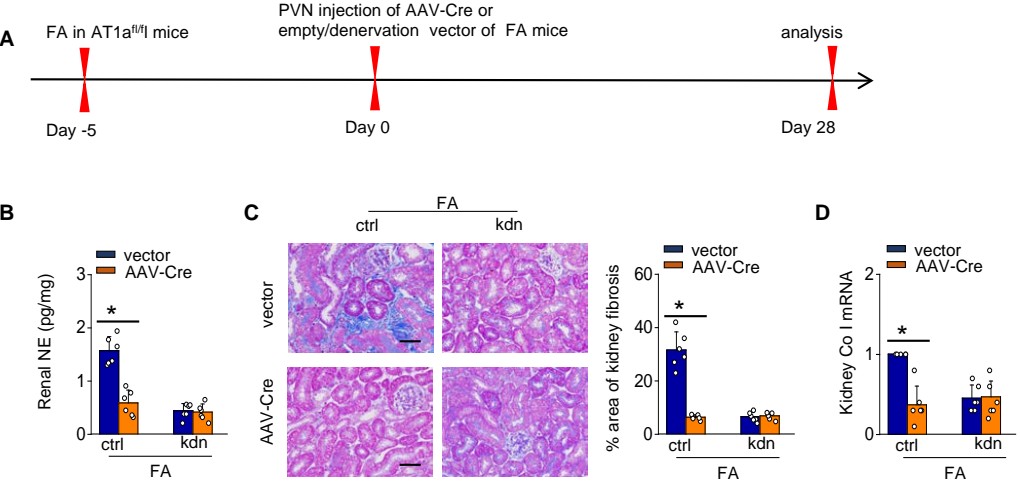

**Figure 4** **Localized Ang II in the PVN triggers fibrosis in the FA-CKD model *via* SNS activation.** Outline of experimental design: PVN-specific deletion of Ang II type 1a receptor (AT1a) was achieved by injecting AAV-Cre into PVN of AT1a$^{fl/fl}$ mice. Blockade of sympathetic outflow was achieved by denervation of the post-obstructed kidney (kdn). (B) Kidney norepinephrine (NE) level in homogenates of the FA-CKD mice. (C) Kidney fibrosis was visualized by Masson trichrome staining in FA-CKD mice: representative images (left) and quantitative data (right). Scale bar, 50 um. (D) Level of collagen I (Co I) mRNA in kidney homogenates of the FA-CKD mice. Error bars, mean $\pm$ standard deviation ($n = 6$ in each group). *, $p < 0.01$. One-way analysis of variance or t test with Bonferroni correction.

We further investigated the mechanism by which activated AT1aR signaling in PVN leads to an increased kidney SNS activity (Fig. 5A). PVN contains excitatory neurons that project to the RVLM, a region involved in regulating SNS outflow. To explore this pathway, we labeled PVN neurons projecting to the RVLM by injecting the RVLM with a CTB-555 (Fig. 5B). In the FA-CKD model, we observed an increase in the signaling from PVN to RVLM, and at the same time, the expression of tyrosine hydroxylase (TH) in the pre-sympathetic neurons of RVLM, which regulate sympathetic nervous system output, was also activated (Figs. 5B–5D). However, by deleting AT1a from PVN, we found that it was possible to reduce the signal transmission from PVN to RVLM and decrease the expression of TH in RVLM (Figs. 5B–5D). This suggests that by blocking the action of AT1a in PVN, it is possible to decrease the PVN-RVLM signal transmission, thereby reducing the central sympathetic activation of the renal sympathetic nerves and exerting a positive impact on the FA-CKD model.

## DISCUSSION

Our main discovery is that the AT1aR signaling in the PVN triggers the activation of a neural pathway that plays a vital role in determining the outcomes following kidney injury induced by folic acid (depicted in Fig. 6). FA-CKD mice activate AT1aR signaling in the PVN neurons projecting to RVLM. Consequently, this activation leads to an increase in SNA in the kidney, ultimately resulting in kidney fibrosis. However, when AT1a in PVN is deleted, the SNS discharge is reduced, and the fibrotic changes in the injured kidney are

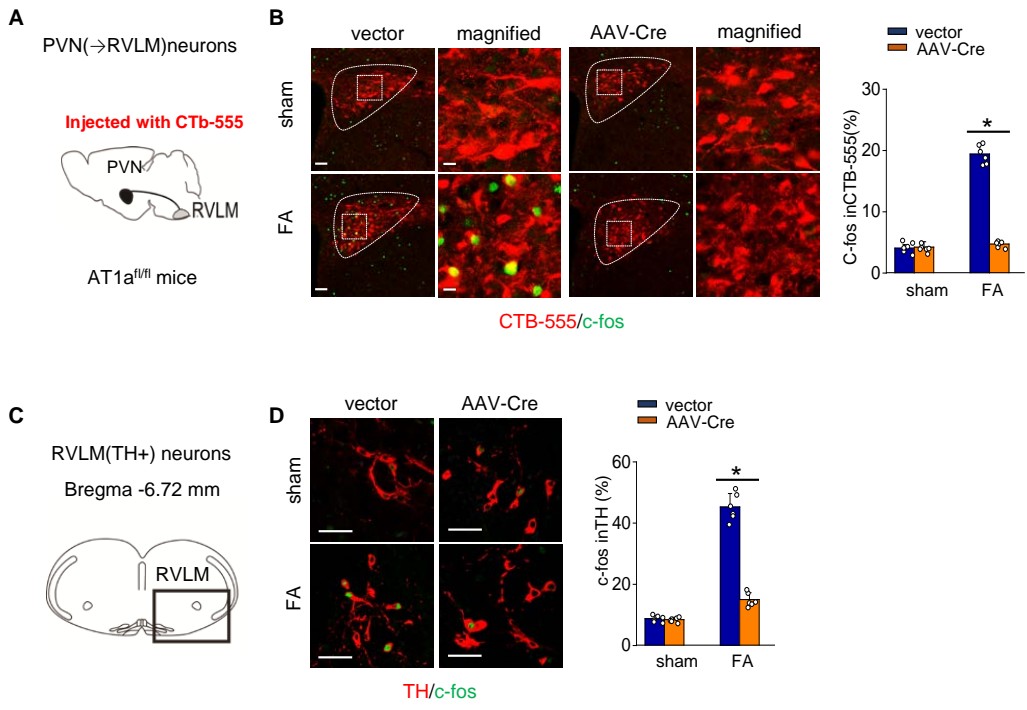

**Figure 5** **Ang II signaling in PVN activates the PVN-RVLM pathway that enhances SNS discharge to the kidney.** (A) Schematic showing the injection of a retrograde tracer, CTb-555, into the RVLM and retrograde labeling of PVN (→RVLM) neurons. The source for the component is from *Nomura et al. (2019)*. (B) Expression of c-fos was detected in CTb-555 PVN neurons: representative images and summary of percentage of c-fos+ cell in CTb-555+ cells. Scale bar, 50 um. *, $p < 0.01$. (C) Schematic of the coronal brain section indicating the RVLM. (D) Immunostaining of c-fos and tyrosine hydroxylase (TH) in RVLM: representative images (left) and summary of percentage of c-fos+ cell in TH+ cells(right). Scale bar, 50 um. *, $p < 0.01$.

reversed. These findings underscore the importance of the PVN-RVLM neural pathway as a significant contributor to the development of kidney fibrosis after folic acid-induced kidney injury. Furthermore, these results suggest the possibility of targeting the PVN neural to prevent the development of kidney fibrotic response in patients.

After folic acid-induced kidney injury, the severity of injury in the acute phase is a highly effective indicator of progression to CKD (*Chawla et al., 2011*). However, the underlying mechanisms that facilitate renal fibrosis after drug toxic challenge remain to be elucidated. Previous studies have found that the upregulation of brain AT1aR signaling is associated with kidney injury induced by 5/6-nephrectomy (*Cao et al., 2015*). However, there have limited knowledge about which precise brain regions are most affected by AT1aR signaling after kidney injury, signal switches in the brain that modulate the sympathetic discharge ,and how this affects the outcomes of the kidneys. In this previous study, we utilize the FA-CKD model to study the brain in relation to renal fibrosis caused by nephrotoxicity. We focus on the PVN, which is strongly implicated as a key site for neural integration (*Guyenet, 2006*). The result of the study showed that while the expression of AT1a in the PVN did not increase in the FA-CKD model, there was an elevation in the levels of AGT

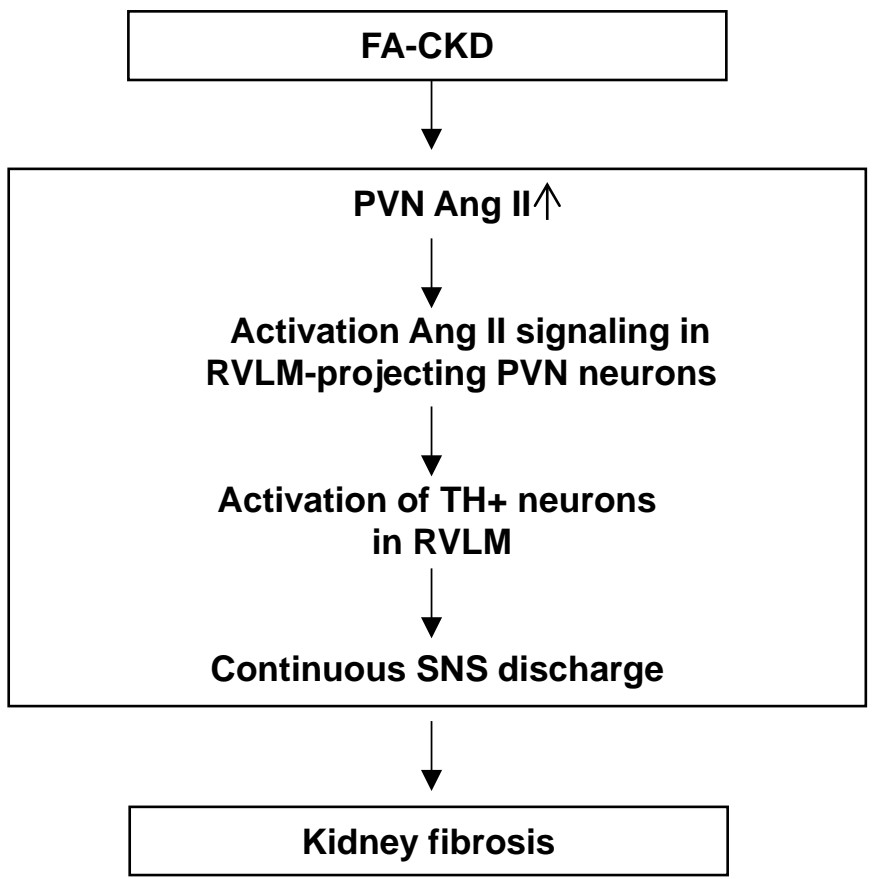

**Figure 6** **The schematic diagram summarizes the steps that link an PVN AngII to renal fibrosis in FA-induced CKD mice.** Folic acid-induced CKD mice activate Ang II signaling in PVN neurons projecting to RVLM, thereby driving SNS discharge and exacerbating renal fibrosis.

and Ang II in the PVN. Through these analyses, we found a strong correlation between the PVN RAS activation and a fibrotic renal outcome after folic acid-induced kidney injury. Specific removal of AT1a expression in the PVN reduced renal fibrosis. These results suggest that the PVN renin-angiotensin system plays a significant role in the folic acid-induced nephrotoxicity mouse model.

After kidney injury, neural involvement has been implicated as a critical contributor to renal fibrosis, and the excessive activation of the sympathetic nervous system promotes renal fibrosis (*Cao et al., 2023*; *Jang, Kim & Padanilam, 2019*; *Tanaka & Okusa, 2020*). Our comprehensive investigation has revealed that the activation of the RAS in the PVN plays a vital role in stimulating PVN(→RVLM) neurons, thereby contributing to the development of renal fibrosis induced by FA. Following kidney injury, the activation of PVN(→RVLM) neurons occurs, leading to an increase in the number of TH-positive neurons in the RVLM. This activation leads to an increase in the SNS outflow that is known to drive organ fibrosis. Our study showed that the activation of kidney SNS. Consistent with previous studies, the levels of NE, a neurotransmitter in the kidney, were found to be elevated. The heightened

release of NE from renal sympathetic nerves through alpha-2 adrenergic receptors ($\alpha$2-AR) contributes to the development and progression of CKD. In the FA-CKD model, the targeted deletion of AT1a specifically in the PVN prevents the projection of PVN $\rightarrow$ RVLM neurons, which rescues the kidney SNS and fibrosis. Therefore, the activation of the PVN RAS following kidney injury selectively stimulates parvocellular neurons in this region *via* AT1a, subsequently increasing their projections to the RVLM, which in turn enhances peripheral SNA, ultimately resulting in the progression of CKD.

Currently, there are no clinical strategies available to prevent the loss of renal function caused by kidney damage following toxic exposure. This study offers valuable insights into the neural mechanisms that regulate renal outcomes in the FA-CKD model and presents evidence supporting several potential strategies that warrant consideration to address this therapeutic gap. One such strategy involves utilizing radiofrequency ablation to disrupt the renal nerves. In mouse models of ureteral obstruction peripheral blockade of $\alpha$2-adrenergic receptors reduces renal sympathetic nerve activity (*Su et al., 2023*). Therefore, blocking alpha-2 adrenergic receptors could effectively interrupt the SNS pathway. Interfering with the action of AT1a receptors using angiotensin receptor blockers or inhibiting Ang II production *via* angiotensin-converting enzyme inhibitors would lead to disruption of the central pathway. Furthermore, the administration of centrally acting sympathetic drugs could mitigate excessive SNS activity. It should be noted that further investigation is necessary; however, these findings hold promise for the development of potential therapeutic approaches in treating patients with kidney damage.

The study has some limitations. We have not investigated how the central PVN RAS system is activated. Previous studies have suggested that in the 5/6 nephrectomy rat model, the damaged kidney activates the PVN RAS *via* renal afferent nerves (*Cao et al., 2017*). Second, this study only employed FA induced renal injury, other models of renal injury induced by nephrotoxic agents need further validation. Third, It's crucial to note that while males have a higher prevalence of renal fibrosis and CKD compared to females, this does not diminish the importance of studying both sexes. In subsequent studies, we will conduct experiments with equal sex ratios to ensure a comprehensive understanding of the disease's mechanisms.

## CONCLUSION

In conclusion, these studies have discovered a neural pathway that plays a crucial role in determining the development of renal fibrosis after folic acid-induced kidney injury. Folic acid-induced CKD mice activate AT1aR signaling in PVN neurons. Subsequently, this activation stimulates AT1a-positive PVN neurons that project to the RVLM, where SNS activity is engaged to initiate fibrotic processes in the FA-CKD model.

### Funding

This study was supported by the National Natural Science Foundation of China (grant number. 82100710), the Guangdong Basic and Applied Basic Research Foundation (grant number.2020A1515110398), the Shenzhen Science and Technology Program (grant number. RCBS20210609103234061), the Shenzhen Key Medical Discipline Construction Fund (grant number. SZXK009) and the Sanming Project of Medicine in Shenzhen(grant number. SZSM202211013). The funders had no role in study design, data collection and analysis, decision to publish, or preparation of the manuscript. The funders had no role in study design, data collection and analysis, decision to publish, or preparation of the manuscript.

### Grant Disclosures

The following grant information was disclosed by the authors:
National Natural Science Foundation of China: 82100710.
Guangdong Basic and Applied Basic Research Foundation: 2020A1515110398.
Shenzhen Science and Technology Program: RCBS20210609103234061.
Shenzhen Key Medical Discipline Construction Fund: SZXK009.
Sanming Project of Medicine in Shenzhen: SZSM202211013.

### Competing Interests

The authors declare there are no competing interests.

### Author Contributions

- Qijun Wan conceived and designed the experiments, performed the experiments, analyzed the data, prepared figures and/or tables, authored or reviewed drafts of the article, and approved the final draft.
- Zhichen Yang performed the experiments, analyzed the data, prepared figures and/or tables, and approved the final draft.
- Lingzhi Li performed the experiments, prepared figures and/or tables, and approved the final draft.
- Liling Wu conceived and designed the experiments, authored or reviewed drafts of the article, and approved the final draft.

### Animal Ethics

The following information was supplied relating to ethical approvals (i.e., approving body and any reference numbers):

All experimental procedures were approved by animal Ethics Committee of Nanfang Hospital and the Animal Ethics Committee of Shenzhen Zhongke Industrial Holdings Co., and were conducted in compliance with their guidelines.

### Data Availability

The raw data is available in the Supplementary File.

## Supplemental Information

Supplemental information for this article can be found online at http://dx.doi.org/10.7717/peerj.18166#supplemental-information.

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
