# Peer review of "Central Angiotensin II type 1 receptor deficiency alleviates renal fibrosis by reducing sympathetic nerve discharge in nephrotoxic folic acid–induced chronic kidney disease"

_PeerJ, doi:10.7717/peerj.18166_

## Round 0.1 · original submission · Major Revisions

Please address concerns of all reviewers and amend manuscript accordingly.

Reviewer 1 ·

Basic reporting

The study identifies a critical neural pathway involving PVN-mediated sympathetic activation that drives renal fibrosis post-folic acid-induced kidney injury, suggesting new avenues for therapeutic intervention. However, some important topics need to be better described.

Experimental design

- It is not Ang II signaling, it should be AT1R signaling. Please fix that in the text.
- For the virus injection, please clarify the volume used and attach the image showing the solution was in the PVN and not affecting regions around the PVN or RVLM.
- How much tissue was used to determine the Ang II levels? How was the tissue collected?
- Check figure 5B and D, they look the same however the excel data shows different numbers.
- Justify the use of male mice in this study.
- Line 76: Clarify the FA operation.
- How was collagen quantified?
- Where is the antibody validation for Ang II?
- What was the signal/noise and the threshold used to determine the quantification Ang II levels?

Validity of the findings

The findings are considered novel and meaningful however the study design needs to be clarified in order to validate the findings.

Additional comments

no comments

·

Basic reporting

The manuscript is well-structured, and the experimental design is sound. The study's strengths include the comprehensive investigation of neural pathways, targeted genetic deletions, and the relevance of the findings to potential clinical applications. However, the manuscript could benefit from addressing a few limitations, such as the mechanisms of central PVN RAS activation and validation in other nephrotoxic injury models.

Experimental design

sound

Validity of the findings

The authors provide a thorough analysis and robust data supporting the correlation between PVN RAS activation and renal fibrosis in the FA-CKD model. The findings align well with previous studies linking brain Ang II signaling with kidney injury, advancing our understanding of the specific brain regions and neural mechanisms involved.

Additional comments

I am pleased to report that the study presents significant and novel findings regarding the role of Ang II signaling in the PVN and its impact on renal fibrosis following folic acid-induced kidney injury. The research offers valuable insights into the neural mechanisms that regulate renal outcomes in CKD and highlights potential therapeutic strategies.

It's crucial to note that while males have a higher prevalence of renal fibrosis and chronic kidney disease (CKD) compared to females, this does not diminish the importance of studying both sexes. The protective effects of estrogen in females and the detrimental effects of androgens in males are significant factors. Post-menopausal women, despite being less frequently affected, have an increased risk of developing renal fibrosis due to the decline in estrogen levels. Therefore, it is advisable to conduct experiments with equal sex ratios to ensure a comprehensive understanding of the disease's mechanisms.

The study convincingly demonstrates that Ang II signaling in the PVN activates a neural pathway involving PVN neurons projecting to the RVLM, which increases sympathetic nerve activity (SNA) in the kidney and leads to fibrosis. Significantly, the targeted deletion of AT1a in the PVN reduces SNS discharge and reverses fibrotic changes, underscoring the critical role of the PVN-RVLM pathway in the development of kidney fibrosis post-injury.
In conclusion, this study significantly contributes to the field and provides a solid foundation for future research and therapeutic development. I recommend the manuscript for publication, subject to minor revisions addressing the aforementioned limitations.

·

Basic reporting

In the manuscript the authors have identified the role of the PVN-renal axis in regulating renal fibrosis after folic acid-induced kidney fibrosis.
Overall Impression: The manuscript is well-written but contains some grammatical and spelling errors, such as:
• Line 39: "failure occurs"
• Line 96: "to deletion of AT1"
• Line 103: "vg/ml"
• Line 183: comparisonsn

Specific Comments:
• The term "operation" for IP injection is misleading, suggesting surgery was performed (Line 76).
• Check section thickness units for staining in all methods and figure legends (Lines 133, 147, etc.).
• There are two GAPDH sequences in the primer sequence list; Please clarify.
• There is a contradiction in the results where kidney denervation is said to abolish the protective effects of PVN AT1a deficiency, but the results indicate both have similar effects. Can you please verify?

As the authors themselves have suggested further studies are required in other models of renal injury as well as to elucidate how the central PVN RAS system is activated.

Experimental design

No Comment

Validity of the findings

No Comment

---

## Round 0.2 · accepted · Accept

All issues pointed b the reviewers were adequately addressed and revised manuscript is acceptable now.

Reviewer 1 ·

Basic reporting

The authors addressed all the concerns.

Experimental design

N/A

Validity of the findings

na

Additional comments

na